# Assessment of Enamel Color Stability of Resins Infiltration Treatment in Human Teeth: A Systematic Review

**DOI:** 10.3390/ijerph191811269

**Published:** 2022-09-07

**Authors:** Matteo Saccucci, Denise Corridore, Gabriele Di Carlo, Elisa Bonucci, Marco Cicciù, Iole Vozza

**Affiliations:** 1Department of Oral and Maxillofacial Sciences, Sapienza University of Rome, 00185 Rome, Italy; 2Department of Biomedical and Dental Sciences and Morphological and Functional Imaging, Messina University, 98100 Messina, Italy

**Keywords:** resin infiltration, ICON, color stability

## Abstract

(1) The evolution of techniques and materials used in dentistry has led to the introduction of a technique known as micro-infiltration, using ICON infiltrating resin. The purpose of this study was to evaluate whether the resin infiltrant can remain stable in the enamel color of human teeth over time or if it causes discoloration and review current knowledge on color stability based on the literature selected solely on studies performed on human teeth and to provide a perspective on the methods proposed by clinicians in the infiltration procedure; (2) Methods: This systematic review was performed according to the Preferred Reporting Items for Systematic Reviews and Meta-Analyses (PRISMA) statement; (3) Results: Twelve studies were selected for this review. The study results suggest that the device content is sufficiently comprehensive. The reviewers expressed strong support for the device’s content for assessing the quality of reviews. The paper summarizes current reports regarding the color stability assessment of enamel treated by in- filtration resin confirmed in in vitro and in vivo studies; (4) Conclusions: Based on these considerations, the resin infiltration method can be recommended to improve the appearance of enamel lesions. The infiltrated lesions remained chromatically stable, showing no significant color changes in the long term.

## 1. Introduction

The evolution of techniques and materials used in dentistry has led to the introduction of a technique known as micro-infiltration, using ICON infiltrating resin (Icon, DMG, Hamburg, Germany) [1]. This technique was first introduced to halt the progression of initial non-cavitated carious lesions by blocking out the penetration of cariogenic acids, thus promoting the preservation of healthy tooth tissue [2]. Its use was rapidly extended to improve the appearance of discoloration, stains resulting from fluorosis, white spots, and alterations caused by trauma or genetic imperfections (amelogenesis imperfecta) affecting tooth enamel formation.

This technique, known as micro-infiltration, is minimally invasive, painless, and therefore particularly advantageous for anxious patients. It allows the management of lesions in a single session and enables the treatment of cavities extending from the outer third to the inner third of the enamel.

The material used is a low-viscosity, fluid, methacrylate resin, mainly triethylene glycol dimethacrylate (TEGDMA). Its penetration coefficient is 147 cm/s, and its refraction coefficient is 1.52. The resin is infiltrated into the porosity of the enamel and fills the empty areas between the hydroxyapatite crystals in a capillary fashion. The air and water trapped in the enamel porosity have a lower light refraction index than that of the healthy tooth. This causes unsightly discoloration ranging from white in the case of white spots to yellow/brown in the case of fluorosis or brown spots. Healthy, normal, saliva-moisturized enamel has a refractive index of 1.62, while demineralized enamel has a refractive index of 1.00 to 1.33. By treating the defect with resin infiltration, the enamel acquires an index of 1.52, which is very close to that of healthy enamel, camouflaging the lesion by optically aligning it with the rest of the healthy enamel surface and resulting in a more aesthetic smile [3,4,5]. Recent studies have questioned the color stability of ICON resin-treated areas, as they appear to be easily subject to extrinsic pigmentation. This could be a problem in the long-term compromising the aesthetic result. The infiltration technique is less invasive than micro abrasion or restoration treatment with composite resins [6], and resin infiltration is able to penetrate deep into the lesions, much more so than the remineralization effect of fluorides.

Both active and inactive enamel lesions can be treated, although infiltration has shown better results in active lesions [7].

On the other hand, the use of ICON infiltration resin is contraindicated in the case of deep lesions, which exceed the outer third of the enamel.

The aim of this study was therefore to assess whether the resin infiltrant can remain stable in enamel color of human teeth over time or whether it can become discolored and review current knowledge of the stability of the color based on the selected literature. In this review, only studies carried out on human teeth were selected and the results of color stability over time and susceptibility to chromogenic agents in the in vivo studies compared with in vitro were evaluated, highlighting the criticalities that the latter present.

## 2. Materials and Methods

In order to identify relevant studies investigating recent scientific evidence regarding the chromatic stability of infiltrating resins, bibliographic research was carried out. The present systematic review followed the PRISMA guidelines (Moher, D. et al., 2009). A PICO (Population, Intervention, Comparison, and Outcome) question was utilized to formulate a focused question and guide the inclusion and exclusion criteria of the present systematic review.

### 2.1. Research Strategy

The literature review was performed by identifying the main research question through which the current literature is investigated: what is the recent scientific evidence regarding the chromatic stability of resin infiltration? What are the strengths and weaknesses of these services? Subsequently inclusion and exclusion criteria were identified. In this systematic review, we included studies that assessed the chromatic stability of resin infiltration and were published in English between 2013 and January 2022. The review involved a systematic search of the journal indexing databases PUBMED, SCOPUS, Google Scholar, and Web of Science. The inclusion criteria were as follows: original articles made on human teeth included cross-sectional, cohort, case-controlled studies, randomized controlled trials, and non-randomized clinical trials regarding the type of study. The exclusion criteria were as follows: publications in a language other than English, no full text available, letters to the editor, editorials, book chapters, interviews, and studies that did not answer the question of this review. Systematic reviews were considered exclusively for the cross-references.

### 2.2. Keywords and Search String

The relevant keywords identified to describe the matter of study are the following: “Tooth Discoloration”, “White Spot Lesion”, “Infiltrant Resin”, “Color Stability”, “Dental Materials”, “Spectrophotometry”, “Icon Infiltrant”, and “Esthetic Restorative Materials”.

The first group of words “Tooth Discoloration”, “White Spot Lesion”, relates to the target population, i.e., people with discoloration or changes to the enamel surface.

The second group of terms “Infiltrant Resin”, “Color Stability”, “Spectrophotometry”, “Icon Infiltrant”, concerns the scope and type of intervention evaluated on the population.

Finally, combining the two individual search strings described above with the Boolean operator “AND” results in the final search string: (“Tooth Discoloration, White Spot Lesion”) AND (“Infiltrant Resin” OR “Color Stability” OR “Icon Infiltrant” OR).

### 2.3. Quality Analysis

The Study Quality Assessment Tools model was used to assess the internal validity of a study using National Heart, Lung, and Blood Institute and Research Triangle Institute International for Observational Cohort and Cross-Sectional Studies and Study Quality Assessment Tools Guidance for Assessing the Quality of Controlled Intervention Studies. Critical evaluation involves the presence of potential selection bias, information bias, measurement bias, or confounding. Examples of confounding include co-interventions, baseline differences in patient characteristics, and other problems during previous questions. In general terms, a “good” study has the least risk of bias, and results are valid. A “fair” study is susceptible to some bias deemed not sufficient to invalidate its results. A “poor” rating indicates significant risk of bias. The high risk of bias results in a poor-quality evaluation. A low risk of bias translates into a good quality assessment (Table 1).

## 3. Results

The Preferred Reporting Items for Systematic Reviews and Meta-Analyses (PRISMA) flow diagram for study selection is displayed in Figure 1.

The first phase of the process consists of identifying the search string and its application to the database mentioned above, inserting the various methodological filters and the inclusion/exclusion criteria. In this first phase, 44 articles were identified. In the second phase, the projection one, after reading the abstract of the 44 identified articles, 27 articles were excluded (17 remaining). In addition, a detailed analysis of the bibliography was carried out, which led to the inclusion of 3 additional articles that may have been excluded from the bibliographic search, for a total of 20 articles. After reading the full text, 8 articles were excluded because they did not meet the inclusion criteria, ultimately identifying 13 articles. If there was a discrepancy in any of the article selections, two of the co-authors discussed the reasons for discrepancies and reached a consensus. In the third phase, the admissibility one, no article was excluded. The remaining 12 articles were carefully read by the two different reviewers in order to verify the actual relevance to the research question and to the inclusion and exclusion criteria (Figure 1). The following information was extracted for final presentation in Table 2: Year and Title, Author and County, Study Population and main Results, Assessment measure, and Aim and Conclusion. The main features of the study methodology are summarized in Table 3.

## 4. Discussion

Enamel infiltrated with ICON resin appears to be more susceptible to staining than untreated enamel areas. From the studies analyzed, it emerged that the color changes were mainly due to the composition of the Icon infiltration resin, which consists mainly of TEGDMA, a monomer that has the ability to infiltrate deep into the lesion [20] and is highly hydrophilic. Due to these characteristics, water absorption in the resin increases [21,22], and color stability is hindered [23]. The change in color, in fact, is favored by the absorption of water, which transports the pigments deep into the resin [24]; furthermore, if the resin material is capable of absorbing water, it could also absorb other liquids, with consequent alteration of the color [16,25,26].

In addition to the characteristics of the resin, color change is also influenced by the patient’s habits, oral hygiene, and diet [27]. To increase the longevity of resin infiltration in aesthetically relevant areas, patients should avoid consuming colored drinks and foods. In particular, they should avoid the consumption of red wine as, according to some studies [9,10,28], it showed the highest average staining value, followed by coffee and tea. Red wine has the highest staining potential due to its alcohol content and low pH. The absorption of alcohol molecules into the resin matrix causes a softening of the polymeric material [29,30] and facilitates the adsorption of pigments on the surface, especially the tannins in which it is rich, thus contributing to coloration [31,32].

Considering color as a complex phenomenon, several factors were considered that could influence the general perception of tooth color, such as lighting conditions, translucency, opacity, light scattering, and the human eye [28].

In order to eliminate potential subjective errors in the assessment of color, spectrophotometers were used in the studies analyzed, which allow an objective assessment and provide precise quantitative data. The spectrophotometer measures the spectral distribution of light and transforms it into color values or numerical values using the CIE L*a*b* system, also known as CIELAB, developed in 1978 by the Commission Internationale de l’éclairage (CIE) [27]. In this system, the value L* indicates brightness (L + = brightness and L− = darkness and varies from 0 = black to 100 = white), the coordinate a* represents the red/green axis (a* + = redness and a* − = green), and the coordinate b* represents the yellow/blue axis (b* + = yellow and b* − = blue). The values of the coordinates a* and b*, when approaching zero, indicate neutral colors (white and grey). The total color change is indicated by the parameter (ΔE) [27]. This technique was chosen to assess color change (ΔE) because it is suitable for determining even the smallest changes [33].

The currently available clinical data on long-term color stability, assessed with a spectrophotometer, have reported stable infiltrates and aesthetic assimilation of color and brightness differences between the infiltrated lesions and the adjacent healthy enamel [12,18,19]. However, if a color change should occur, polishing or whitening treatment of the infiltrated area can be carried out. Studies have shown that polishing infiltrated lesions increases their resistance to staining and can minimize the staining effect [8,14,15,33,34]. In fact, polishing has resulted in a significant reduction in ΔE values, and this is since dyes are absorbed on the surface, with little penetration into resin materials or the dental substrate [19,35].

As an alternative to polishing the surfaces, a bleaching treatment can be carried out. The ΔE values obtained after whitening of Icon-infiltrated surfaces showed a color change similar to that obtained in healthy enamel. These results indicate that it is possible to increase the brightness of Icon-infiltrated enamel, making the use of polishing procedures unnecessary.

Limitations of the present systematic review were the variability among the studies and the heterogeneity in various parameters such as the time interval taken into consideration and the methods of evaluating color stability.

## 5. Conclusions

In conclusion, within the limitation of this systematic review, the results of in vitro tests indicate that the infiltrated enamel can undergo color changes under the influence of chromogenic agents. If these variations occur, it is possible to minimize them with the aid of polishing or bleaching techniques. In in vivo test infiltrated lesions remained chromatically stable, showing no significant color changes in the long term. To ensure the long-term success of the infiltrating resin treatment, further in vivo tests are required to assess the long-term color stability and the risk of discoloration or pigmentation more accurately.

## Figures and Tables

**Figure 1 ijerph-19-11269-f001:**
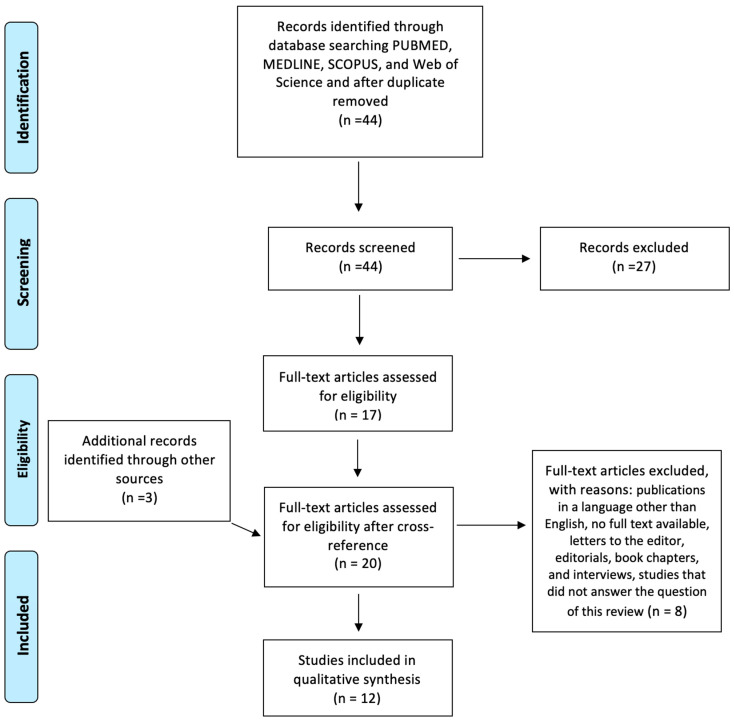
PRISMA diagram for research strategy.

**Table 1 ijerph-19-11269-t001:** Study quality assessment. CD, cannot determine; NA, not applicable; NR, not reported; Y, yes; N, no.

Year and Title	References		Was the Study Question or Objective Clearly Stated?	Was the Study Population Clearly and Fully Described, Including a Case Definition?	Were the Cases Consecutive?	Were the Subjects Comparable	Was the Intervention Clearly Described?	Were the Outcome Measures Clearly Defined, Valid, Reliable, and Implemented Consistently Across All Study Participants?	Was the Length of Follow-Up Adequate?	Were the Statistical Methods Well-Described?	Were the Results Well-Described?
In-vivo color stability of enamel after ICON^®^ treatment at 6 years of follow-up: a prospective single-center study. J Dent. 2022 Jan 13:103943. doi: 10.1016/j.jdent.2021.103943.	[8]	GOOD	Y	Y	Y	Y	Y	Y	Y	Y	Y
Color stability of resin infiltrated white spot lesion after exposure to stain-causing drinks. Saudi Journal of Biological Sciences (2021), doi: 10.1016/j.sjbs.2021.09.063	[9]	GOOD	Y	Y	Y	Y	Y	Y	Y	Y	Y
Effect of tea on color stability of enamel lesions treated with resin infiltrant Dentistry, 18, art. no. 4. (2021) doi: 10.18502/fid.v18i4.5434	[10]	GOOD	Y	Y	Y	Y	Y	Y	Y	Y	Y
Esthetic Management of Incisors with Diffuse and Demarcated Opacities: 24 Month Follow-up Case Report Operative Dentistry, 45 (6), pp. 569–574. (2020) doi: 10.2341/19-206-S	[11]	POOR	Y	Y	NA	NA	Y	NA	Y	NA	Y
Resin Infiltration in Dental Fluorosis Treatment—1-Year Follow-Up *Medicina* 2021, 57, 22. (2020) doi: 10.3390/medicina57010022	[12]	GOOD	Y	Y	Y	Y	Y	Y	Y	Y	Y
Long-term follow-up of camouflage effects following resin infiltration of post orthodontic white-spot lesions in vivo. Angle Orthod 89: 33–39. (2019) doi: 10.2319/052118-383.1	[13]	GOOD	Y	Y	Y	Y	Y	Y	Y	Y	Y
Surface properties and color stability of resin-infiltrated enamel lesions Operative Dentistry, 41 (6), pp. 617–626 (2016) doi: 10.2341/15-319-L	[14]	GOOD	Y	Y	Y	Y	Y	Y	Y	Y	Y
Evaluation of staining and color changes of a resin infiltration system Angle Orthodontist, 86 (6), pp. 900–904. (2016)	[15]	GOOD	Y	Y	Y	Y	Y	Y	Y	Y	Y
Hydrolytic and color stability of resin infiltration: a preliminary in vitro trial. J Contemp Dent Pract 7: 377–381. (2016) doi: 10.5005/jp-journals-10024-1858	[16]	GOOD	Y	Y	Y	Y	Y	Y	Y	Y	Y
Evaluation of stain penetration by beverages in demineralized enamel treated with resin infiltration. Oper Dent; 41: 93–102. (2016) doi: 10.2341/13-259-L	[17]	GOOD	Y	Y	Y	Y	Y	Y	Y	Y	Y
Resin infiltration for aesthetic improvement of mild to moderate fluorosis: A six-month follow-up case report Oral Health and Preventive Dentistry, 13 (4), pp. 317–322. (2015) doi: 10.3290/j.ohpd.a32785	[18]	POOR	Y	Y	NA	NA	Y	NA	Y	NA	Y
Camouflage effects following resin infiltration of post orthodontic white-spot lesions in vivo: one-year follow-up. Angle Orthod; 85: 374–380. (2015) doi: 10.2319/050914-334.1	[19]	GOOD	Y	Y	Y	Y	Y	Y	Y	Y	Y

**Table 2 ijerph-19-11269-t002:** Summary table of systematic review.

Year and Title	References	Autor and County	Materials and Methods	Aim and Conclusion
In-vivo color stability of enamel after ICON^®^ treatment at 6 years of follow-up: a prospective single center study. J Dent. Jan 13:103943. (2022) doi: 10.1016/j.jdent.2021.103943.	[8]	Mazur M, Westland S, Ndokaj A, Nardi Gm, Guerra F, Ottolenghi L Italy	76 teeth previously treated with ICON^®^ due to hypo mineralized lesions of enamel were recalled for a follow-up. Color stability was assessed: (i) subjectively by patients using FDI-color matching criteria; (ii) objectively by calculating CIEDE2000 color differences between the affected/treated and sound enamel in each tooth at T0 (baseline), T1 (one year) and T2 (six years) based on spectrophotometric data. Analysis of correlation between FDI and CIEDE2000 data was performed.	This in-vivo clinical study provides subjective and objective documentation on color stability of enamel after resin infiltration at a mean observation time of six years after treatment. This study shows that caries infiltration satisfactorily masks aesthetically relevant lesions after longer follow-up. Subjective and objective outcomes showed a fair correlation mainly for the initial masking effect.
Color stability of resin infiltrated white spot lesion after exposure to stain-causing drinks. Saudi Journal of Biological Sciences (2021), doi: 10.1016/j.sjbs.2021.09.063	[9]	Alqahtani, S., Abusaq, A., Alghamdi, M., Shokair, N., Albounni, R Saudi Arabia	Thirty-three extracted human premolar teeth were used to create WSLs, and ICON resin infiltration treatment was performed to obliterate the enamel pores. Teeth with resin infiltrated WSLs were sectioned into two halves by cutting mesiodistally and cross-sectionally at 1 mm below the CEJ. The resin infiltrated specimens were exposed to control (artificial saliva) and staining subgroups. Color stability was assessed using the CIE L*a*b* system.	Assess the discoloration effect of red tea, Arabic coffee, and black coffee on the resin infiltrated white spot lesions (WSL). Moreover, to investigate the impact of time (1d, 3d, and 7d) on the discoloration of the resin infiltrated WSLs. Resin infiltrated WSLs showed marked color changes after exposure to red tea, black coffee, and Arabic coffee over time. Severe discoloration of the infiltrant was evident with the use of red tea compared to black coffee and Arabic coffee. This suggests that ICON resin-based composite material might not be a suitable material for WSL infiltration.
Effect of tea on color stability of enamel lesions treated with resin infiltrant Dentistry, 18, art. no. 4. (2021) doi: 10.18502/fid.v18i4.5434	[10]	Arjomand, M.E., Ganjkar, M.H., Ghamari, R. Iran	This in vitro, experimental study evaluated 30 extracted human third molars with no caries, cracks, or enamel defects. The samples were divided into three groups of 10, namely sound enamel, demineralized enamel, and demineralized enamel plus RI. White spot lesions (WSLs) were artificially created. Next, Icon RI was applied on the samples in group 3. The baseline color of the samples was measured using a spectrophotometer. They were immersed in tea solution 3 times a day, each time for 15 min, for a period of 2 weeks and then underwent colorimetry again. Data were analyzed using one-way ANOVA.	This study aimed to assess the effect of tea on color stability of enamel lesions treated with resin infiltrant (RI). Within the limitations of this in vitro study; the results showed that tea solution caused clinically unacceptable color change in all groups. However, the color stability of WSLs treated with RI was significantly lower than other groups following immersion in tea solution.
Esthetic Management of Incisors with Diffuse and Demarcated Opacities: 24 Month Follow-up Case Report Operative Dentistry, 45 (6), pp. 569–574. (2020) doi: 10.2341/19-206-S	[11]	dos Santos Athayde, G., Jorge, R.C., Americano, G.C.A., Barja-Fidalgo, F., Soviero, V.M. Brazil	Clinical examination showed diffuse opacities in teeth 7, 8, 9, and 10, which was diagnosed as moderate fluorosis (score 5) based on the Thylstrup and Fejerskov Index for Dental Fluorosis. A yellow demarcated opacity in tooth 9 was diagnosed as MIH, according to the European Academy of Pediatric Dentistry because the first permanent molars were also affected the treatment decision was based on minimally invasive dentistry, using the infiltration technique with low-viscosity resin (Icon) and composite resin (TPH. Dentsply, São Paulo, Brazil) to mask the discolorations.	This clinical case report describes a minimally Invasive approach to mask diffuse and demarcated opacities in permanent anterior teeth using a resin infiltrant (Icon, DMG, Hamburg, Germany) and composite resin restoration. The esthetic treatment based on resin infiltration and composite achieved excellent results with color stability at the 24-month follow-up. Diffuse and demarcated opacities were masked in a single session that improved the patient’s self-esteem.
Resin Infiltration in Dental Fluorosis Treatment—1-Year Follow-Up Medicina 2021, 57, 22. (2020) doi.org/10.3390/medicina57010022	[12]	Francesca Zotti, Luca Albertini, Nicolò Tomizioli, Giorgia Capocasale and Massimo Albanese Italy	200 fluorosis lesions were treated using ICON infiltrating resin (DMG, Hamburg, Germany). Parameters related to patients were collected by a questionnaire and analyzed aesthetic dissatisfaction about lesions, Shiff Air Index Sensitive Scale, sensitive teeth after treatment, the satisfaction of duration of treatment. The same operator measured dimensions of lesions Tooth Surface Index of Fluorosis (TSIF) and numbers of etching cycles needed for treating lesions. Statistical analysis was performed. The follow-up was 1 year after measurement was performed at baseline (t0), immediately after the treatment (t1), and every three months during the observation period	Regarding aesthetic dissatisfaction for white spots lesions, a significant statistical difference was noticed between values at t0 and t1, as well as for those between t1 and t2. This aspect is of great interest, it indicates that result obtained at the end of the treatment improved during the first three months of follow-up and patients answered more positively regarding the aesthetic results. This improvement was no longer highlighted after the third month of observation, likely meaning that the result of infiltration becomes stable, and it is not subject to changes, at least about patient perception.
Long-term follow-up of camouflage effects following resin infiltration of post orthodontic white- spot lesions in vivo. Angle Orthod 89: 33–39. (2019) doi: 10.2319/052118-383.1	[13]	Knösel M, Eckstein A, Helms HJ. Germany	Of twenty subjects who received previous resin infiltration treatment non-cavitated postorthodontic WSL after multibracket treatment during a randomized controlled trial and were contacted 20 months after baseline, CIE-L*a*b* differences between summarized color and lightness values (DEWSL/SAE) of WSL and SAE were assessed using a spectrophotometer and compared to baseline data assessed prior to infiltration (T0), and those after 6 (T6), and 12 (T12) months using paired t tests at a significance level of a ¼ 5%.	To reassess the long-term camouflage effects of resin infiltration (Icon, DMG, Hamburg, Germany) of white spot lesions (WSL) and sound adjacent enamel (SAE) achieved in a previous trial. Assimilation of infiltrated WSL to the color of adjacent enamel by resin infiltration is considered to be suitable for the long-term improvement in the esthetic appearance of postorthodontic WSL.
Surface properties and color stability of resin-infiltrated enamel lesions Operative Dentistry, 41 (6), pp. 617–626 (2016) doi: 10.2341/15-319-L	[14]	Zhao, X., Ren, Y.-F. Cina United States	Surface area roughness (Sa), Vickers microhardness (VHN), and CIE L∗a∗b∗color values were measured on sound enamel, resin-infiltrated lesions, and untreated lesions before and after aging.	To examine the surface topographies, microhardness, and color stability of resin-infiltrated enamel lesions before and after aging challenges in vitro using three-dimensional laser scanning profilometry, surface microhardness testing, spectrophotometry, and scanning electron microscopy. Surface hardness of enamel lesions increased significantly after resin infiltration and remained stable following thermocycling. Surface roughness and color stability of resin-infiltrated enamel lesions were less than ideal and might further deteriorate after aging in the oral environment
Evaluation of staining and color changes of a resin infiltration system Angle Orthodontist, 86 (6), pp. 900–904. (2016)	[15]	Leland, A., Akyalcin, S., English, J.D., Tufekci, E., Paravina, R. United States	Six groups were formed (n 8 teeth/group) and were exposed to the following: red wine, coffee, orange juice, combined staining agents, accelerated aging, and distilled water for 1 week. The teeth were then polished with a prophy cup and polishing paste. Color properties were assessed using a spectrophotometer at baseline (T0), after each exposure (T1), and after polishing (T2) Color difference (DE∗) was calculated between each time point for both halves of the teeth (RI and NRI). Data were analyzed with a two-way analysis of variance with presence of resin infiltration and staining agents as the main effects for each time point pair. Multiple comparisons were made with a Bonferroni post hoc test. The level of significance was set at P, 0.05.	To analyze the staining and color changes of a resin infiltrant system used for management of white spot lesions (WSLs) RI areas showed higher staining susceptibility than did NRI areas. However, prophylaxis had a strong effect on reversing the discoloration of both RI and NRI areas.
Hydrolytic and color stability of resin infiltration: a preliminary in vitro trial. J Contemp Dent Pract 7: 377–381. (2016) doi: 10.5005/jp-journals-10024-1858	[16]	Ritwik P, Jones CM, Fan Y, Sarkar NK.	Color was recorded by spectral colorimeter. The samples were subjected to four experimental conditions: (1) group 1: Stored in lactic acid solution (pH 4.9) for 24 h; (2) group 2: Thermocycled for 100 cycles (temperatures: 5 °C, 55 °C, and dwell time of 15 s); (3) group 3: Stored in 0.1N sodium hydroxide solution (pH 12.48) for 14 days at 60 °C; (4) group 4: Stored in phosphate-buffered saline solution (pH 7.2) at 37 °C for 4 months. The weight and color were recorded again after removal of the samples from the experimental conditions. Scanning electron microscopy imaging was performed for samples from groups 1, 3, and 4.	This study evaluated the in vitro hydrolytic and color stability of the ICON^®^ resin infiltration system (IC) in 42 extracted human teeth. Infiltration system exhibited greatest weight loss and color change in demineralizing solution.
Evaluation of stain penetration by beverages in demineralized enamel treated with resin infiltration. Oper Dent; 41: 93–102. (2016) doi: 10.2341/13-259-L	[17]	Lee J, Chen JW, Omar S, Kwan SR, Meharry M. USA	Sixty extracted human permanent molars were demineralized, treated with resin infiltration (Icon), and immersed in four different beverages (coffee, grape juice, iced tea, and distilled water; N = 15) for four weeks. After aging, teeth in the distilled water group were stained with 2% methylene blue for 24 h. All teeth were sectioned, and stain penetration was evaluated under light microscopy.	To evaluate stain penetration by different beverages in artificially demineralized human teeth treated with resin infiltration. Both Icon and control surfaces exhibit stain penetration by different beverages (iced tea, grape juice, and coffee). However, resin-infiltrated enamel surfaces allow significantly less depth of stain penetration compared with untreated surfaces. The iced tea group presents greatest depth of stain penetration, followed by grape juice, methylene blue, and coffee
Resin infiltration for aesthetic improvement of mild to moderate fluorosis: A six-month follow-up case report Oral Health and Preventive Dentistry, 13 (4), pp. 317–322. (2015) doi: 10.3290/j.ohpd.a32785	[18]	Auschill, T.M., Schmidt, K.E., Arweiler, N.B. Germany	A 24-year-old woman with corresponds to scores 2 and 3 of Dean’s Fluorosis Index, as more than 25% and less than 50% of each tooth was affected and some brown staining was evident on the fluorosed portions of the teeth. Treatment of the fluorosed areas of the teeth using resin infiltration began two months following the last bleaching procedure of the teeth. the outcome of the treatment was effective and had long-term stability six months after treatment.	To determine whether fluorosed areas of teeth can be successfully treated with resin infiltration and whether the results are long lasting. This case report demonstrates that resin infiltration is an agreeable option for this type of tooth discoloration, rather than choosing more invasive, conventional procedures. More studies need to be completed to determine longer-term outcomes of the technique
Camouflage effects following resin infiltration of post orthodontic white-spot lesions in vivo: one-year follow-up. Angle Orthod; 85: 374–380. (2015) doi: 10.2319/050914-334.1	[19]	Eckstein A, Helms HJ, Knösel M. Germany	Twenty subjects (trial teeth n teeth 5 111) who had received resin infiltration treatment of non-cavitated post orthodontic WSLs were contacted for a 1-year follow-up assessment of CIEL*a*b* colors (T12). Color and lightness (CIE-L*a*b*) data for WSLs and SAE were compared to baseline data assessed before infiltration (T0) and those assessed after 6 months (T6), using a spectrophotometer. The target parameter was the difference between the summarized color and lightness values (DEWSL/SAE). Intergroup (WSL, SAE) and inter-time comparisons (T0 vs. T6, T12) were performed using paired t-tests at a significance level of a 5.5%.	To assess camouflage effects by concealment of post orthodontic white-spot lesions (WSLs) to sound adjacent enamel (SAE) achieved over 12 months with resin infiltration (Icon, DMG, Hamburg, Germany). As color and lightness characteristics of the Icon infiltrant as well as the esthetic camouflage effects achieved by WSL infiltration were not altered significantly or clinically relevant after 12 months, the method of resin infiltration can be recommended for an enduring esthetic improvement of post orthodontic WSL

**Table 3 ijerph-19-11269-t003:** Summary table of systematic review.

Year and Title	References	Autor and County	Study Designed	Follow-Up	Color Stability Assesed
In-vivo color stability of enamel after ICON^®^ treatment at 6 years of follow-up: a prospective single center study. J Dent. Jan 13:103943. (2022) doi: 10.1016/j.jdent.2021.103943.	[8]	Mazur M, Westland S, Ndokaj A, Nardi Gm, Guerra F, Ottolenghi L Italy	follow-up of patients treated with infiltration during a single-center, split-mouth controlled trial	T0 (baseline), T1 (one year) T2 (six years)	Subjectively by patients using FDI-color matching criteria. Objectively by calculating CIEDE2000 color differences between the affected/treated and sound enamel based on spectrophotometric data.
Color stability of resin infiltrated white spot lesion after exposure to stain-causing drinks. Saudi Journal of Biological Sciences (2021), doi: 10.1016/j.sjbs.2021.09.063	[9]	Alqahtani, S., Abusaq, A., Alghamdi, M., Shokair, N., Albounni, R Saudi Arabia	in vitro, experimental study	T0 (baseline), T1 24 h T2 72 h T3 1 week	Color stability was assessed using the CIE L*a*b* system.
Effect of tea on color stability of enamel lesions treated with resin infiltrant Dentistry, 18, art. no. 4. (2021) doi: 10.18502/fid.v18i4.5434	[10]	Arjomand, M.E., Ganjkar, M.H., Ghamari, R. Iran	in vitro, experimental study	T0 (baseline), T1 2week	Color stability was assessed using the CIE L*a*b* system.
Esthetic Management of Incisors with Diffuse and Demarcated Opacities: 24 Month Follow-up Case Report Operative Dentistry, 45 (6), pp. 569–574. (2020) doi: 10.2341/19-206-S	[11]	dos Santos Athayde, G., Jorge, R.C., Americano, G.C.A., Barja-Fidalgo, F., Soviero, V.M. Brazil	clinical case report	T0 (baseline), T1 6 months T2 24 months	Visual valutation
Resin Infiltration in Dental Fluorosis Treatment—1-Year Follow-Up Medicina 2021, 57, 22. (2020) doi.org/10.3390/medicina57010022	[12]	Francesca Zotti, Luca Albertini, Nicolò Tomizioli, Giorgia Capocasale and Massimo Albanese Italy	in vivo experimental study	t0: before the treatment t1: after treatment every 3 months during the observation period (t2, t3, t4)	Effective measurement of main length of lesions was determined by a digital software elaboration (Rasband, W.S., ImageJ, U. S. National Institutes of Health, Bethesda, Maryland, USA) of photographs by the same operator.
Long-term follow-up of camouflage effects following resin infiltration of post orthodontic white- spot lesions in vivo. Angle Orthod 89: 33–39. (2019) doi: 10.2319/052118-383.1	[13]	Knösel M, Eckstein A, Helms HJ. Germany	second follow-up of the patients treated with WSL infiltration during a single-center, split-mouth controlled simple-randomized trial.	T0 (baseline), T1 6 months T2 12 months T3 24 months	Color stability was assessed using the CIE L*a*b* system.
Surface properties and color stability of resin-infiltrated enamel lesions Operative Dentistry, 41 (6), pp. 617–626 (2016) doi: 10.2341/15-319-L	[14]	Zhao, X., Ren, Y.-F. Cina United States	in vivo, experimental study	T0 1 week	Color stability was assessed using the CIE L*a*b* system.
Evaluation of staining and color changes of a resin infiltration system Angle Orthodontist, 86 (6), pp. 900–904. (2016)	[15]	Leland, A., Akyalcin, S., English, J.D., Tufekci, E., Paravina, R. United States	Case control in vitro study	T0 1 week baseline T1 after staining/aging. T2, a after polishing;	Color stability was assessed using the CIE L*a*b* system.
Hydrolytic and colorsta-bility of resin infiltration: a preliminary in vitro trial. J Contemp Dent Pract 7: 377–381. (2016) doi: 10.5005/jp-journals-10024-1858	[16]	Ritwik P, Jones CM, Fan Y, Sarkar NK.	in vitro, experimental study	Group 1 in a demineralizing solution for 24 h at room temperature. Group 2 comprised teeth that were subjected to 100 cycles of thermocycling with a dwell time of 15 s. Group 3 comprised teeth stored in sodium hydroxide solution for 14 days. Group 4 comprised teeth immersed in phosphate-buffered saline for 4 months.	Color stability was assessed using the CIE L*a*b* system.
Evaluation of stain penetration by beverages in demineralized enamel treated with resin infiltration. Oper Dent; 41: 93–102. (2016) doi: 10.2341/13-259-L	[17]	Lee J, Chen JW, Omar S, Kwan SR, Meharry M. USA	in vitro, experimental study	T0 4 weeks. T1 6 months aging T2 in the distilled water group were stained with 2% methylene blue for 24 h.	A light microscope, at a magnification of 303 (Opto-metric Tools, Inc, Rockleigh, NJ, USA) connected to a magnescale LH10 (Sony, Tokyo, Japan), was used.
Resin infiltration for aesthetic improvement of mild to moderate fluorosis: A six-month follow-up case report Oral Health and Preventive Dentistry, 13 (4), pp. 317–322. (2015) doi: 10.3290/j.ohpd.a32785	[18]	Auschill, T.M., Schmidt, K.E., Arweiler, N.B. Germany	Case report	T0 baseline T1 six months after treatment.	Subjective evaluation
Camouflage effects following resin infiltration of post orthodontic white-spot lesions in vivo: one-year follow-up. Angle Orthod; 85: 374–380. (2015) doi: 10.2319/050914-334.1	[19]	Eckstein A, Helms HJ, Knösel M. Germany	follow-up of patients treated with WSL infiltration during a single-center, split-mouth controlled simple-randomized trial	T0 baseline T1 6 months, T2 12 months	Color stability was assessed using the CIE L*a*b* system.

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
