# Peer review of "Assessment of Enamel Color Stability of Resins Infiltration Treatment in Human Teeth: A Systematic Review"

_ijerph, 2022, doi:10.3390/ijerph191811269_

Round 1

Reviewer 1 Report

Dear authors,

Your article is informative, extensive, up-to-date. The information is valuable from a practical point of view. I highly recommend publishing the article as soon as possible, as dentists would benefit from it. Keep up the good work!

Author Response

We thank the reviewer for having thoroughly examined our manuscript. Thank you for your consideration on the topic of our study.

Reviewer 2 Report

The manuscript has been rewritten very well and includes enough citations to support the achievements. In my opinion, the present article can be published in the Journal after text editing.

Author Response

We thank the reviewer for having thoroughly examined our manuscript. Thank you for your consideration on the topic of our study. 

A text editing was carried out

Reviewer 3 Report

The authors performed a systematic review about colour stability over time of infiltrating resin. The article is valid and of clinical interest. I would recommend a professional english editing.

I would suggest the following minor revisions:

Abstract: I would suggest to provide a stronger rationale for your study;

The result section is very poor, please provide a better description of your result, so that the reader could understand what you found right from the abstract.

Introduction: "ICON" is a commercial name, please provide details.

Line 49 "This could be a problem in the long term due to the imperceptibility of the treatment," This sentence is not very clear, please rephrase it.

Again, I suggest to provide a stronger rationale for this systematic review rather than "to assess if". Are there contradictory findings in the literature? Are the existing study homogeneous or not? Why do we need a systematic review? Does another review already exist?

Materials and methods: The first paragraph is a bit confusing: PICO is not a standard for a bibliographic research, but to formulate a research question. Similarly, while PRISMA is a standard for reporting the methods of a SR. Please rephrase.

Lines 78-80 should be reprhased to be clearer for the readers.

It is not clear to me whether the Authors used an existing and pre-validated quality assessment tool or if they developed a custom QAT for their review.

Who performed the quality assessment?

Results: Lines 2-3 are unnecessary;

The reasons for excluding an article should be reported, also in figure 1 (which is showing formatting symbols, please revise). The sentence at line 11-12 should be rephrased, it is not fluent.

The Authors report that two authors extracted data from the finally included articles: how potential discrepancies between the two authors were addressed?

Discussion: Line 29 there is an unformatted citation;

The qualitative synthesis of the included article is missing.

The limitation section is missing.

Conclusions: The first sentence is not appropriate, as you didn't evaluate if the procedure can be recommended, overall (is it clinically effective? Does the cost/benefit ratio favours the procedure? Etc..)....but only if the colour is stable over time. Please stick to the findings of your review.
